# Direct synthesis of partially ethoxylated branched polyethylenimine from ethanolamine

Claire N. Brodie[1], Alister S. Goodfellow [1], Matthew J. Andrews [1], Aniekan E. Owen[1], Michael Bühl[1] ✉ & Amit Kumar [1] ✉

We report here a method to make a branched and partially ethoxylated polyethyleneimine derivative directly from ethanolamine. The polymerization reaction is catalysed by a pincer complex of Earth-abundant metal, manganese, and produces water as the only byproduct. Industrial processes to produce polyethyleneimines involve the transformation of ethanolamine to a highly toxic chemical, aziridine, by an energy-intensive/waste-generating process followed by the ring-opening polymerization of aziridine. The reported method bypasses the need to produce a highly toxic intermediate and presents advantages over the current state-of-the-art. We propose that the polymerization process follows a hydrogen borrowing pathway that involves (a) dehydrogenation of ethanolamine to form 2-aminoacetaldehyde, (b) dehydrative coupling of 2-aminoacetaldehyde with ethanolamine to form an imine derivative, and (c) subsequent hydrogenation of imine derivative to form alkylated amines.

Branched polyethylenimine (PEI) and polyethylenimine ethoxylated (PEIE) with an annual global market size of around £400 million have applications in various areas ranging from adhesives, textiles, cosmetics, water treatment, and detergents[1]. Lately, their applications in areas such as gene delivery[2], tissue culture[3], $CO_2$ capture[4], solar cells[5], and optoelectronic devices[6] have also emerged[3]. The current industrial process for the production of branched polyethylenimine is carried out through acid-catalysed cationic ring opening polymerization of aziridine, which is a highly toxic, mutagenic, and volatile liquid[7]. The hazardous nature of aziridine adds to the production cost as it requires highly efficient scrubbers on vent lines and gasket materials that can withstand prolonged contact with aziridine[1]. Only two companies– BASF (Germany) and Nippon Shokubai (Japan) produce aziridine, both transforming produced aziridine to much less toxic polyethylenimine. BASF currently produces aziridine by the Wenker process in which ethanolamine is first reacted with sulphuric acid at 100–200 °C to give 2-aminoethyl hydrogensulfate that is subsequently reacted with aqueous sodium hydroxide to produce aziridine (Fig. 1A)[1]. The two-step process produces a stoichiometric amount of sodium sulphate waste. Nippon Shokubai produces aziridine by the catalytic dehydration of

ethanolamine at 350–450 °C under reduced pressure (Fig. 1A)[1]. Although the process doesn't produce any waste, it needs a high energy system and multistage distillation setup to separate aziridine from the by-product water. The branched polyethylenimines (PEI) produced using these processes are further reacted with ethylene oxide to make polyethylenimine ethoxylated (PEIE). A process that can bypass the need to make aziridine and produce branched polyethylenimine directly from ethanolamine will be of significant benefit to the environment and economy.

Acceptorless dehydrogenative coupling is an atom-economic approach to make various organic compounds such as nitriles, and carbonyl derivatives[8–10]. The dehydrogenative coupling of alcohols and amines to make imines has been reported using pincer catalysts of ruthenium[11,12], iron[13], and manganese[14,15]. The N-alkylation of amines using alcohols through a hydrogen borrowing mechanism has also been reported using various transition metal complexes[16–18]. The concept of acceptorless dehydrogenative catalysis has also been expanded to make polymers such as polyesters[19], nylons[20,21], and polyureas[22–25]. The synthesis of polyalkylenimines from the coupling of diols and diamines or alkanolamines using ruthenium or iridium

[1]EaStCHEM, School of Chemistry, University of St Andrews, North Haugh, St Andrews KY16 9ST, UK. ✉e-mail: buehl@st-andrews.ac.uk; ak336@st-andrews.ac.uk

**Fig. 1 | Synthesis of polyethylenimines.** Current industrial processes (**A**) for the synthesis of branched polyethylenimine (PEI) and the process reported herein (**B**). Inset shows the structure of the precatalyst used in this study.

catalysts has been claimed in patents filed by the BASF[26,27]. Along this direction, we have recently reported the synthesis of a branched polyethylenimine derivative from the coupling of ethylene glycol and ethylenediamine using the Earth-abundant manganese pincer precatalyst, **Mn-1**, in the presence of KO$^t$Bu, generating the active amido-Mn species, **Mn-2**, in situ. We now report that the same complex efficiently catalyses the direct synthesis of a branched and partially ethoxylated polyethylenimine from ethanolamine through a borrowing hydrogen mechanism (Fig. 1B).

## Results and discussion
### Polymerization optimization
We started our investigation by conducting the optimization of ethanolamine polymerization reaction using the manganese pincer complex **1** by varying base, temperature, solvent, reaction time, and catalytic loading. Remarkably, refluxing ethanolamine (2 mmol) in toluene (4 mL) using 1 mol% complex **Mn-1**, 10 mol% KO$^t$Bu at 150 °C for 24 h in a sealed vessel resulted in the isolation of a solid product in 81% yield (Table 1; Entry 1; reproduced three times). This water-soluble product, that precipitates during the reaction course, was determined to have a high molecular weight ($M_n$ 38,700 g mol$^{-1}$) and reasonably narrow polydispersity (Đ 1.21) by GPC (Gel Permeation Chromatography). An IR (ATR-FTIR) spectrum obtained of the product exhibits stretching frequencies at v2934 cm$^{-1}$ and v1584 cm$^{-1}$, corresponding to C-H stretch and N-H bend, respectively. The presence of alcohol end groups is indicated by a broad resonance with v3254 cm$^{-1}$ (O-H) and strong bands at v1350 cm$^{-1}$ and v1061 cm$^{-1}$. NMR spectroscopic analysis ($^1$H and $^{13}$C{$^1$H}) of the product in D$_2$O suggests a branched polymeric structure (shown in Fig. 1), with signals corresponding to tertiary ($\delta_C$ 63–58 ppm), secondary ($\delta_C$ 48–52 ppm) and primary ($\delta_C$ 44–40

ppm) amine environments observed within a $^{13}$C{$^1$H} NMR spectrum (Fig. 2A), and broad overlapping resonances observed between ~3.75 and 2.5 ppm within a $^1$H NMR spectrum[28]. Crude quantification to estimate the percentage of primary amine has also been carried out, through reactions to form Ruhemann's purple monitored by UV-vis spectrometry and through quantitative $^{13}$C{$^1$H} NMR spectroscopy, which showed ~27% primary amine (ESI section 1.10).

A comparable yield (78%) can be obtained using THF as reaction solvent (Entry 2), however, the polymer produced in THF is of lower average molecular weight and an, apparently, polymodal distribution is observed within the resulting GPC trace (Supplementary Fig. 16), indicated through increased polydispersity (Đ 1.47). Additionally, the presence of unsaturation (imine functionality) is observed within an IR spectrum (v$_{C=N}$ 1655 cm$^{-1}$) and NMR spectroscopic analysis ($\delta_C$ 163.8 ppm and $\delta_H$ 8.06). Performing the reaction in anisole allowed the preparation of a polymeric sample with similar molecular weight and dispersity to that produced in toluene, now in 72% yield (Entry 3). However, inspection of a $^{13}$C{$^1$H} NMR spectrum of the product (Entry 3) in D$_2$O reveals a degree of unsaturation, i.e., imine, present within the polymer backbones, indicated by the presence of signals with $\delta_C$ 164.7 and 164.0 ppm.

Changing the base from KO$^t$Bu to K$_2$CO$_3$ retains selective formation of **PEI-1**, however with a much diminished yield after 24 h (c.f. 81% and 13%) (Entry 4). A further variation of the base to NaO$^t$Bu resulted in a loss of selectivity, generating a polymodal mixture containing amide ($\delta_C$ 179.4 ppm), imine ($\delta_C$ 164.8 ppm), and enamine ($\delta_C$ ~140 ppm) functionalities (Entry 5).

Decreasing the equivalence of KO$^t$Bu base from 1:10 [**Mn-1**]:[KO$^t$Bu] to 1:5 resulted in a decrease in the yield obtained from 81% to 49% (Entry 6), a fact that we speculate may be due to the

**Table 1 | Reaction condition optimization for ethanolamine polymerization mediated by Mn-1[a]**

| Entry | Cat. /mol % | Base (mol%) | Product(s) | Yield[b] /% | $M_n$[c] /g mol⁻¹ | Đ[e] | $T_g$ /°C |
|---|---|---|---|---|---|---|---|
| 1 | 1 | KO^tBu (10) | PEI-1 | 81 | 38,700 | 1.21 | −8.2 |
| 2[d] | 1 | KO^tBu (10) | u-PEI-1 | 78 | 22,500 | 1.47 | 0.5 |
| 3[e] | 1 | KO^tBu (10) | u-PEI-1 | 72 | 34,500 | 1.20 | −0.9 |
| 4 | 1 | K_2CO_3 (10) | PEI-1 | 13 | 30,100 | 1.26 | — |
| 5 | 1 | NaO^tBu (10) | Mixture | 75 | 31,800 | 1.28 | — |
| 6 | 1 | KO^tBu (5) | u-PEI-1, PEI-1 | 49 | 50,700 | 1.14 | — |
| 7[f] | 1 | KO^tBu (10) | PA, PEI-1 | 29 | 36,400 | 1.21 | −1.1 |
| 8[g] | 1 | KO^tBu (10) | — | — | — | — | — |
| 9[h] | 1 | KO^tBu (10) | PEI-1[i] | 64 | >55,000 | — | — |
| 10[j] | 1 | KO^tBu (10) | PEI-1 | 84 | 40,090 | 1.23 | — |
| 11 | 0.5 | KO^tBu (10) | u-PEI-1 | 52 | 50,000 | 1.08 | −1.3 |
| 12[k] | 1 | KOt^Bu (10) | PEI-1 | 90 | 47,600 | 1.11 | — |
| 13[l] | 1 | KOt^Bu (10) | PEI-1 | 84 | 50,900 | 1.11 | — |
| 14 | — | KOt^Bu (10) | DEA | 37[m] | — | — | — |
| 15 | 1 | — | — | — | — | — | — |
| 16[n] | 1 | KOt^Bu (10) | DEA | 41 | — | — | — |
| 17[o] | 1 | — | PEI-1 | 77 | 33,200 | 1.29 | — |

Where PEI-1 = polyethyleneimine-1; u-PEI-1 = unsaturated PEI-1; PA = polyamide and DEA = diethanolamine.
[a]Experimental conditions: 2 mmol ethanolamine 0.02 M substrate in toluene, 1 mol% **Mn-1**, 150 °C, 24 h, sealed 100 cm³ vessel.
[b]Isolated yield, based on $C_2H_4N$ repeating units.
[c]Determined by GPC.
[d]THF.
[e]Anisole.
[f]120 °C.
[g]Reaction carried out under 20 bar hydrogen gas.
[h]Reaction carried out using the recycled catalyst.
[i]increased degree of ethoxylation.
[j]20 mmol scale, 1 M ethanolamine, 0.5 mol% **Mn-1**, stainless steel autoclave.
[k]0.25 equiv. ethylene diamine added.
[l]0.50 equiv. ethylene diamine added.
[m]Average of 2 repeats.
[n]Mn(CO)_5Br. DEA = diethanolamine.
[o]**Mn-2**.

involvement of KO^tBu in dehydrogenation and dehydration steps. Decreasing the reaction temperature from 150 °C to 120 °C (Entry 7) results in a significantly diminished yield (now 29%), coupled with a loss in selectivity with some amide functionality observed using ¹³C{¹H} NMR spectroscopy ($δ_C$ 179.6 ppm).

The thermal properties of **PEI-1** produced from these reactions were probed through thermogravimetric analysis and differential scanning calorimetry. Where a glass transition ($T_g$) was detected, these were observed between −10 °C and 1 °C, higher than that of commercial PEI samples (c.f. branched-PEI $T_g$ below −40 °C and linear-PEI $T_g$ below −20 °C)[29,30]. However, these values may be in line with what might be the expected range for a PEIE (e.g. $T_g$ − 24 °C for a branched PEIE)[6,31] derived from a linear-PEI, consistent with high degrees of ethoxy end-groups. Complex thermal decomposition with significant mass loss is observed above ~200 °C, with some small mass loss below this temperature also observed which could indicate the presence of residual substrate within the samples despite drying under reduced pressure and elevated temperatures prior to analyses (Supplementary Figs. 69–71). Performing powder X-ray diffraction on a sample of PEI-1 reveals this material to be largely amorphous (Supplementary Fig. 84).

**Scale up and catalyst recyclability**

Performing the reaction in a rigorously cleaned stainless steel autoclave under $H_2$ gas (20 bar) in an attempt to generate a fully saturated product was unsuccessful, likely due to inhibition of the alcohol dehydrogenation step through saturation by dihydrogen activation (Table 1; Entry 8). We have also studied the recyclability of the catalyst (see Supplementary Information, Section 1.5), and although a polymer of high molecular weight >55,000 g mol⁻¹ (Fig. 2B) was obtained in 64% yield (Table 1; Entry 9) using the recycled catalyst, spectroscopic analysis (IR, NMR) suggested an increased degree of ethoxylation at polymer chain ends ($v_{C-O}$ 1051 cm⁻¹)[32]. The thermal analysis of this isolated polymer showed decomposition onset at 200 °C, without passing through a melt phase – suggesting this material is mainly amorphous. The reaction was scaled up to 20 mmol ethanolamine using a substrate concentration of 1 M and 0.5 mol% pre-catalyst **Mn-1**, leading to the production of **PEI-1** in a good yield of 84% (Table 1; Entry 10). The recovered material has spectroscopic data, molecular weight ($M_n$ 40,090 g mol⁻¹), dispersity (Đ 1.23), and thermal characteristics of that using the small-scale optimised conditions (Table 1; Entry 1).

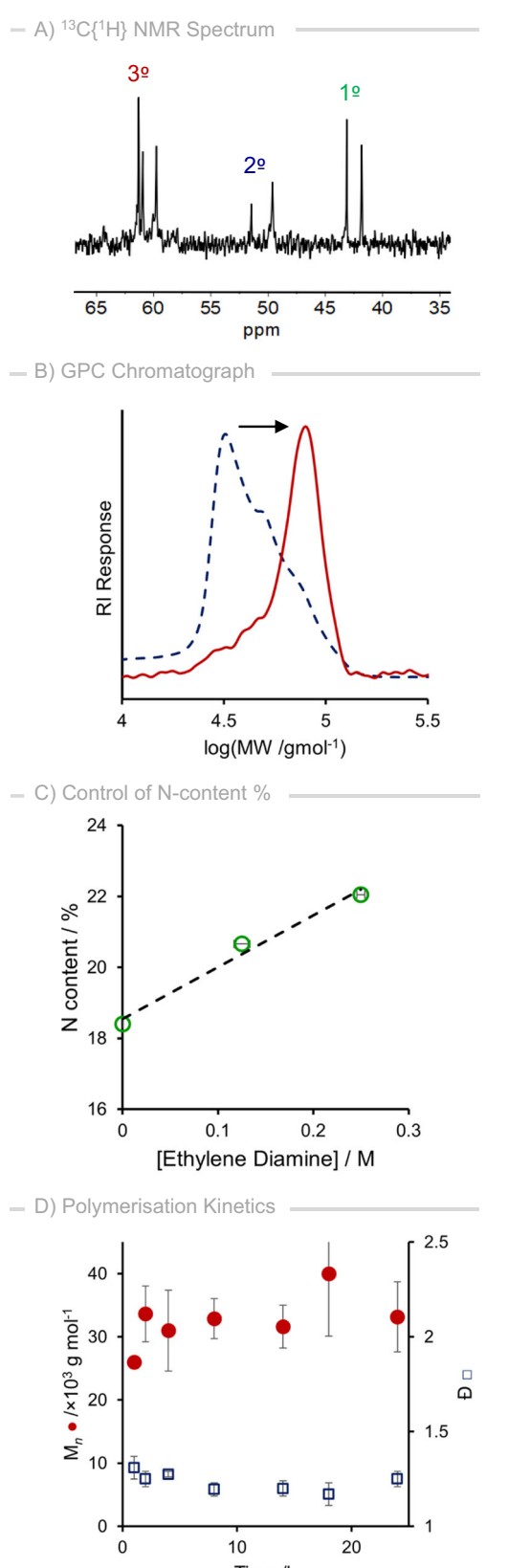

— A) ¹³C{¹H} NMR Spectrum —

3º

2º

1º

65 60 55 50 45 40 35
ppm

— B) GPC Chromatograph —

RI Response

4 4.5 5 5.5
log(MW /gmol⁻¹)

— C) Control of N-content % —

N content / %

[Ethylene Diamine] / M

— D) Polymerisation Kinetics —

Mₙ /×10³ g mol⁻¹

Ð

Time /h

**Fig. 2 | Characterisation details of PEI-1 and polymerisation kinetics. A** ¹³C{¹H} NMR (D₂O, 126 MHz) of **PEI-1**; (**B**) GPC traces of **PEI-1** (dashed) and **PEI-1-E** obtained from catalyst recycling experiment (red solid line); (**C**) Relationship between ethylene diamine concentration and resulting N content; (**D**) Kinetics of polymerization for **PEI-1** synthesis: Mₙ (red circles) and Ð (blue squares) vs reaction time.

## N-content control

The efficiency of **PEI** for several of the specialty applications (e.g. $CO_2$ capture) can depend upon the number of primary amine sites present within the material[4,33,34]. As such, we were interested in gaining control over the degree of ethoxylation present within the **PEI-1** material obtained within this study (Table 1; Entries 1, 13, and 14). To our satisfaction, doping the ethanolamine polymerization reactions with sub-stochiometric quantities of ethylenediamine reveals a linear relationship between [diamine] and measured nitrogen content (Fig. 2C), as determined by elemental analysis (Supplementary Table 3).

A control reaction in the absence of a base returns unreacted starting materials. However, performing the reaction in the absence of pre-catalyst, **Mn-1**, but in the presence of catalytic quantities of KOᵗBu (10 mol%), yields 37% conversion of ethanolamine to *N,N*-diethanolamine. Performing the reaction in the presence of both base and Mn(CO)₅Br (Table 1, Entry 16) returns a similar result, with 41% conversion to diethanolamine. These control experiments suggest that both the precatalyst **Mn-1** and KOᵗBu are needed for the catalytic transformation. Performing the reaction with pre-activated **Mn-2** and the absence of base also yields a polymer in a similar yield (77%) (Table 1, Entry 17), however, the degree of branching was found to be significantly higher with only ~5% primary amine and a greater proportion of tertiary amine (see Supplementary Information sections 1.6.9 and 1.10.2).

**Mechanistic investigations.** From our product characterization, it follows that the dehydrogenative coupling of aminoethanol generates a polyethyleneimine derivative that includes both branched and linear regions with both amino- and alcohol-end groups. In order to obtain some control over this polymerization, we were therefore interested in investigating the mechanism of dehydrogenation and polymerization through a combined experimental and computational approach (DFT computations at the PBE0-D3(BJ)ₚ𝒸ₘ(ₜₕ𝒻)/def2-TZVP//RI-BP86ₚ𝒸ₘ(ₜₕ𝒻)/def2-SVP level which has been validated and used in previous studies)[22,23,35,36]. The overall process can be described as following a hydrogen borrowing mechanism where water is the only by-product with the following fundamental steps: (1) Mn-mediated alcohol dehydrogenation to aminoaldehyde; (2) reaction of aldehyde with additional amine to generate transient hemiaminal; (3) dehydration to produce an intermediate imine or enamine; (4) Mn-mediated imine/enamine hydrogenation resulting in the formation of secondary/tertiary amine repeating units. This overall mechanistic picture is in line with previous DFT studies reported on hydrogen borrowing processes for the N-alkylation of amines by alcohols catalysed by manganese[37-40], cobalt[41], nickel[42], ruthenium[43-45], and iridium[46,47].

## Alcohol dehydrogenation

Having demonstrated the direct synthesis of the branched polyethyleneimine derivative, we first diverted our attention to gain insight into the catalytic cycle involving organometallic intermediates. The stochiometric reaction between complex **Mn-1** and KOᵗBu generates the previously reported activated amido-complex, **Mn-2** (Fig. 3A)[48]. We have previously reported the spontaneous and quantitative conversion of **Mn-2** to alkoxide complex upon reaction with ethylene glycol at room temperature[36]. However, the reaction of ethanolamine with complex **Mn-2** at room temperature resulted in the formation of a mixture of complexes, the predominant species being the anticipated alkoxide complex (**Mn-3**), δ_P 81.3 ppm as 53% of the reaction mixture; calculated to be exergonic ($\Delta G = -1.2$ kcal mol⁻¹).

We have located key transition states for a few selected Mn-catalysed (de)hydrogenation steps. The profile for the dehydrogenation of the substrate ethanolamine (Supplementary Fig. 180) is very similar to that computed for dehydrogenation of ethylene glycol[36] and our calculations predict the barrier for ethanolamine dehydrogenation to be $\Delta^{\ddagger}G = 17.7$ kcal mol⁻¹ (Fig. 3). Restoring the active Mn catalyst

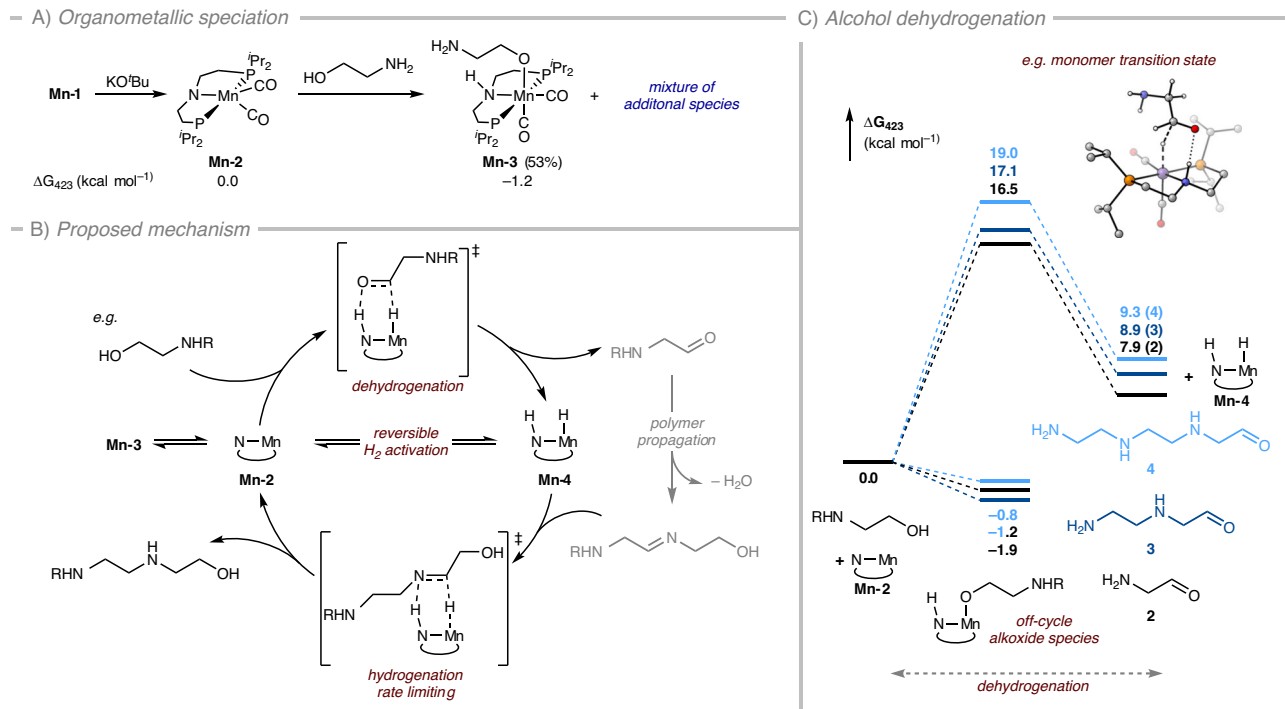

**Fig. 3 | Proposed pathways for the synthesis of polyethylenimines.**
**A** Stoichiometric reactivity between activated complex (**2**) with ethanolamine; (**B**) simplified catalytic mechanism of (de)hydrogenation; (**C**) Energies ($\Delta G_{423,15K}$) of alcohol dehydrogenation for monomer, dimer and trimer (PBE0-D3(BJ)$_{PCM(THF)}$/def2-TZVP//RI-BP86$_{PCM(THF)}$/def2-SVP).

(**Mn-2**) through reversible dihydrogen release (assisted by a monomer proton relay, see Supplementary Fig. 180 in the ESI) increases the overall barrier to 27.3 kcal mol$^{-1}$ [49].

Experimentally, dihydrogen release from Mn-hydride (**Mn-4**) is demonstrated by headspace gas analysis of a dehydrogenative polymerization of ethanolamine under our optimized conditions where the reaction is stopped after 5 h. Gas analysis (GC-TCD) of the mixture by GC-TCD shows the presence of free dihydrogen (Supplementary Fig. 117). Stochiometric evolution of H$_2$ is, however, not observed. In line with a hydrogen borrowing pathway, it is used in the sequential hydrogenation of dehydrated intermediates that are produced from the condensation of aldehydes and amines. With this in mind, we have constructed a reaction profile for the direct transfer hydrogenation in dimer formation, with a barrier higher than H$_2$ release and driven by the exergonic formation of the saturated product ($\Delta^{\ddagger}G = 33.8$ kcal mol$^{-1}$, $\Delta_rG = -3.9$ kcal mol$^{-1}$, Supplementary Fig. 182).

## Imine/Enamine hydrogenation

For the formation of linear and branched **PEI-1**, there is a requirement for the hydrogenation of respective intermediate imine and enamine species. Our calculations suggest the hydrogenation of unsaturated species to be rate limiting, with barriers for the trimer imine and enamine hydrogenation calculated as 21.3 and 36.1 kcal mol$^{-1}$, respectively, relative to the free unsaturated systems and the hydrogenated catalyst (Supplementary Fig. 181). This is similar to a report on the N-alkylation of amines by alcohols catalysed by a bifunctional iridium complex where reduction of imines was proposed as a rate determining step[50].

NMR spectroscopic ($^1$H, $^{13}$C{$^1$H}) monitoring of the polymerization reaction under the catalytic conditions described in Table 1, Entry 1 over the course of the reaction (0.5–24 h) showed increasing concentration of imine moieties up to 8 h and their subsequent disappearance during 8–24 h of the reaction period, supporting that the reaction proceeds via imine intermediates (Supplementary Figs. 97,

98). To probe imine hydrogenation in isolation, we have also demonstrated the hydrogenation of N-benzyl-1-phenylmethanimine to dibenzylamine using complex **Mn-1** (1 mol%) and KO$^t$Bu (10 mol%) in toluene, in line with our optimized catalytic conditions (Table 1; Entry 1) at low hydrogen pressure (1 bar, 298 K). After 24 h under our conditions, there is observed 15% hydrogenation of this model imine substrate (see Supplementary Information section 1.6.4). To probe the possibility of alkene hydrogenation under our conditions, a sample of 1,5-cyclooctadiene (COD) was subjected to our catalytic conditions (1 mol% **Mn-1**, 10 mol% KO$^t$Bu, 150 °C) under H$_2$ (1.2 bar). After 24 h, GC-MS analysis of the resulting mixture revealed 83% conversion to cyclooctane and cyclooctene in a 3:1 selectivity ratio, respectively. (see Supplementary Information section 1.6.5) Similar tests were also carried out using enamine 1-(1-butenyl)pyrrolidine which showed full consumption of the $^1$H NMR peak corresponding to the enamine, however with a complex mixture of unidentifiable products was obtained by $^1$H NMR and GC-MS (see Supplementary Information section 1.6.6).

We have also carried out the polymerisation of ethanolamine under a D$_2$ atmosphere and a $^2$H NMR spectrum of the product shows deuterium incorporation into the polymer backbone (see Supplementary Information section 1.6.8). This is supportive of a hydrogen-borrowing mechanism where there is release of at least some H$_2$ gas that can come back to hydrogenate imine.

## Polymer propagation

GPC chromatographs (relative to monodisperse PEG/PEO standards) taken throughout the time course of the reaction showed the generation of high molecular weight (~30,000 g mol$^{-1}$) material with a narrow polydispersity (Đ <1.2) from the early stages of the reaction (Fig. 2D). These data may suggest that polymerization proceeds via a chain-growth type mechanism. This contrasts with what might be expected for condensation polymerization, which would typically proceed via a step-growth mechanism. Indeed there are some reports

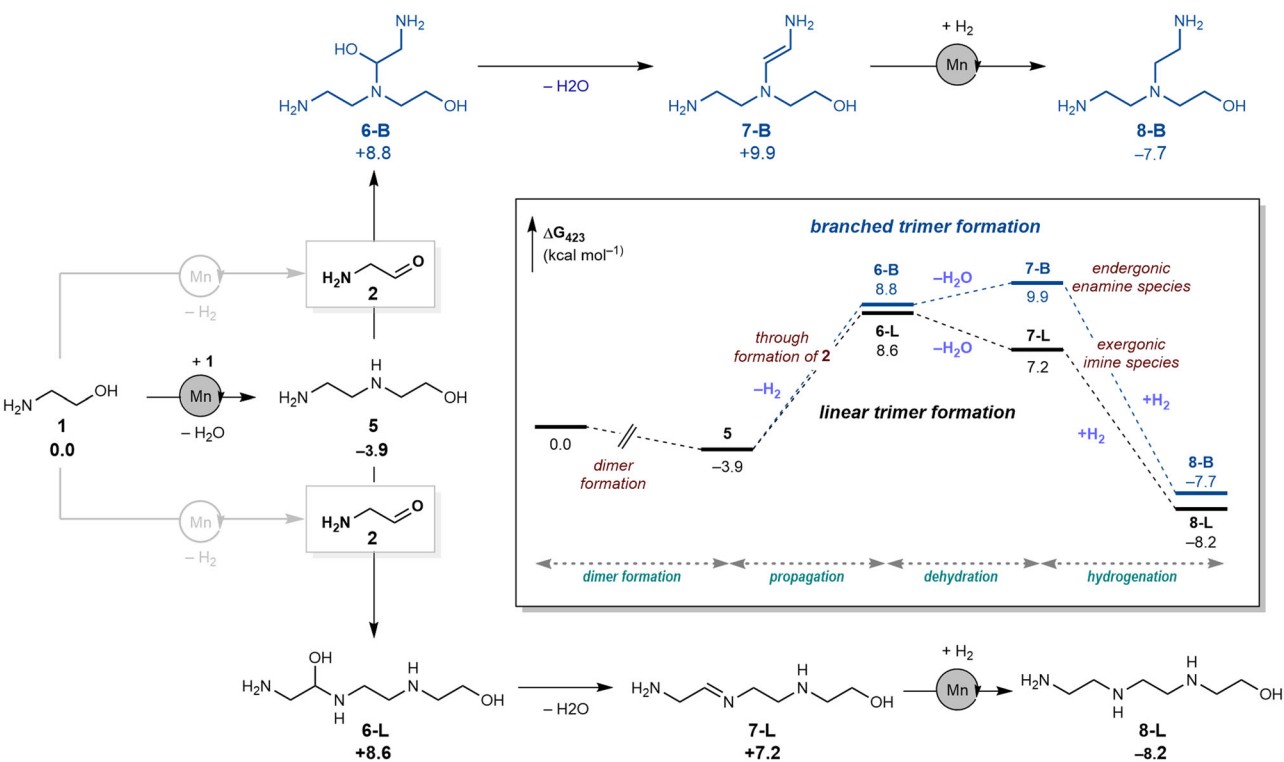

**Fig. 4 | Proposed pathways for the chain propagation.** Possible pathways for the formation of linear and branched **PEI-1** and corresponding thermodynamic driving forces. Computed free energies $\Delta G_{423.15K}$ in kcal mol$^{-1}$ relative to **1** at the PBE0-D3(BJ)$_{PCM(THF)}$/def2-TZVP//RI-BP86$_{PCM(THF)}$/def2-SVP level.

of chain-growth mechanisms in operation for condensation polymerization reported previously[51,52]. For example, due to a substituent effect, the reaction of a monomer with a polymer/oligomer end functional group can be faster than the reaction of a monomer with another monomer leading to the chain growth polymerisation[52]. However, we do note that accurate representation of polymer molecular weight by GPC may be complicated through the possibility of cross-linking, and that the molecular weight obtainable for this process may be limited by phase-separation during the reaction course[53]. Computational consideration of the thermodynamic driving force for the dehydrogenation to form monomer aldehyde (**2**), dimer (**3**) and trimer (**4**) indicates only a slight preference for monomer aldehyde formation where $\Delta\Delta_r G = 1.7$ and 1.0 kcal mol$^{-1}$, respectively (Fig. 3C). We also note that with regeneration of the active catalyst **Mn-2**, the comparative barrier for dehydrogenation to produce **2** is also lower by $\Delta\Delta^{\ddagger}G = 1.7$ and 1.0 kcal mol$^{-1}$, respectively (Supplementary Fig. 165). These computational results do not provide any strong preference for either chain-growth or step-growth mechanism. Thus, based on our studies we are unable to confirm whether the chain propagation occurs via chain-growth or step-growth and it is possible that both the pathways are occurring under the reaction conditions.

NMR spectroscopic analysis of the **PEI-1** produced reveals the presence of primary, secondary and tertiary amine sites, suggesting the possibility of both linear and branched regions within the polymer structure. To produce both linear or branched amines, we propose a common intermediate, the dimer of **1**: *N*-(2-hydroxyethyl)ethylenediamine (**5**, Fig. 4). To test if *N*-(2-hydroxyethyl)ethylenediamine (**5**) is a feasible intermediate in the generation of **PEI-1**, it was used directly as a substrate using the optimised conditions described in Table 1, Entry 1. This reaction produced **PEI-1** in high molecular weight ($M_n$ 46,100 g mol$^{-1}$) and narrow dispersity (Đ 1.22) suggesting that **5** is a possible intermediate in the synthesis of **PEI-1**.

Condensation of *N*-(2-hydroxyethyl)ethylenediamine (**5**) with aminoaldehyde (**2**) can proceed through nucleophilic attack by the

terminal or secondary amine present in **5**, resulting in the generation of a linear or branched hemi-aminal species (**6-L** or **6-B**, respectively, Fig. 4). These have similar relative energies ($\Delta G = 8.6$ and 8.8 kcal mol$^{-1}$); however, these pathways diverge upon consideration of the subsequent dehydration step. Dehydration of the linear species (**6-L**) to produce an intermediate imine (as observed spectroscopically, e.g. **7-L**, see ESI, Supplementary Figs. 88, 89) is predicted to be exergonic by 1.4 kcal mol$^{-1}$, while the same process to produce a branched enamine intermediate (**7-B**) is endergonic by 1.1 kcal mol$^{-1}$. The resulting hydrogenated linear or branched amines generated are of similar relative energy, both possessing intramolecular hydrogen bonds; though, the barrier to hydrogenation is higher for enamine than for imine. Formation of branched polymer is therefore expected to proceed through intermediate enamines, hydrogenation of which is predicted to be overall rate limiting. However, the accumulation of enamines within the reaction mixture would not be expected due to their endergonic formation and the large thermodynamic driving force for subsequent hydrogenation. We believe that the driving forces associated with the dehydration of unsaturated intermediates and the following barriers of hydrogenation rationalise the spectroscopic observation of only imine intermediates (and not enamines) during the reaction course.

## Substrate scope

We were also interested to see whether this dehydrative transformation could be extended to other amino-alcohols. As such, several amino alcohols with varying chain lengths and substitution patterns as described in Fig. 5 were studied to expand the substrate scope for this reaction. Interestingly, high conversion to a polyamine product seems to be limited to ethanolamine. The introduction of a substituent adjacent to the amine functionality, while maintaining the 1,2-amino-alcohol substitution pattern on a C$_2$ backbone, results in the generation of pyrazines and derivatives (Fig. 5). In all cases where the backbone separation of amine and alcohol functionalities was greater than

**Fig. 5 | Substrate scope.** Additional amino alcohols studied for the Mn-catalysed dehydrogenative coupling.

$C_2$, lactam formation was observed. Furthermore, in the case of 5-amino-1-pentanol, there is also observed production of 2,3,4,5-tetrahydropyridine as the major product. Further details on these unselective reactions can be found in the Supplementary Information, section 1.11.

In summary, we report here the direct synthesis of branched partially ethoxylated polyethylenimine derivative (**PEI-1**) from ethanolamine feedstock. The reaction is mediated by a manganese pincer precatalyst **1** (1 mol%) in the presence of KO$^t$Bu (10 mol%) and forms **PEI-1** ($M_n$ 38,700 g mol$^{-1}$, Đ 1.2) in 81% yield. Considering the current energy-intensive (e.g. temperature 350–450 °C) and waste-generating methods used to produce a highly toxic intermediate aziridine (from ethanolamine) that is needed to produce polyethylenimine, the method described herein uses much milder conditions, avoids toxic intermediates, and is highly atom-economic. The isolated polymers have been characterized by NMR and IR spectroscopy, GPC, elemental analysis, TGA, and DSC studies. Control over the relative amine and alcohol content is demonstrated through doping the polymerization with terminating ethylenediamine. Combined experimental and computational mechanistic studies of the process suggest that the polymerization follows a hydrogen-borrowing mechanism involving manganese-catalysed dehydrogenation of ethanolamine to form an aminoaldehyde followed by a condensation process to form a hemiaminal intermediate during chain propagation. The hemiaminal intermediate then dehydrates, generating intermediate imine or enamine species that subsequently undergo hydrogenation catalysed by manganese, ultimately leading to the formation of a branched polyethyleneimine (**PEI-1**). (De)hydrogenation steps are predicted to occur via an outer-sphere metal-ligand cooperative mechanism. Dehydration reactions affording imines are slightly exergonic, whereas dehydration reactions affording isomeric olefins tend to be slightly endergonic and lead to the formation of branched oligomers. Model linear or branched oligomer (e.g. trimer) have similar relative energies, supporting the formation of branched polyethylenimine derivative (**PEI-1**).

## Methods
### General methods
See Supplementary Information for synthetic details, polymerisation optimisation, spectra, control studies and computational details.

### Optimised polymerisation method
A 100 mL ampoule equipped with a J-Young's valve was charged with pre-catalyst (0.02 mmol, 1 mol%) and KO$^t$Bu (0.20 mmol, 10 mol%). Toluene (4 mL) and ethanolamine (2.0 mmol) were added, and the flask was sealed under an argon atmosphere before heating (150 °C) for 24 h with stirring (400 rpm). After this period, the reaction vessel was allowed to cool to room temperature. The product was extracted into distilled water (5 mL) and any volatile components were removed under reduced pressure at 110 °C.

## Data availability
All data generated or analysed during this study are included in this published article (and its Supplementary Information file). The research data supporting this publication can be accessed at https://doi.org/10.17630/b31cd4a0-ad35-439d-a14c-e5a9e1ad4d48. All data are available from the corresponding author upon request. Source data (coordinates of the optimized structures) are provided. Source data are provided with this paper.

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

## Acknowledgements

This research is funded by a UKRI Future Leaders Fellowship (MR/W007460/1) and an EPSRC grant (EP/Y005449/1). A.S.G thanks the EaSI-CAT centre for Doctoral Training for funding. M.B. wishes to thank the School of Chemistry and EaStCHEM for their support. A.E.O. gratefully acknowledges a fellowship from the Akwa Ibom State University (TET-Fund). Calculations were performed on a local computer cluster maintained by Dr. H. Früchtl. We thank Dr. Julia Payne for her help with the powder-XRD study and James Luk with reproducibility controls.

## Author contributions

A.K. conceived the study. The synthetic chemistry was carried out by C.N.B. and M.J.A. A.S.G., A.E.O. and M.B. performed D.F.T. computation.

Data analysis was carried out by all authors. The paper was written and prepared by C.N.B., A.S.G., M.J.A., M.B., and A.K.

## Competing interests

The authors declare no competing interests.

## Additional information

**Peer review information** : *Nature Communications* thanks the anonymous reviewers for their contribution to the peer review of this work. A peer review file is available.

