## [Peer Review File · Nature Communications]

Direct Synthesis of Partially Ethoxylated Branched Polyethylenimine from EthanolamineReviewers' Comments:

Reviewer #1:

Remarks to the Author:

This manuscript reports a very interesting and new approach to obtain highly valuable and versatile ethoxylated polyamines. It uses a simple and common starting material (ethanolamine) in combination an earth-abundant metal (manganese) based catalyst and therefore this approach could have a high potential for later applications. I could allow access to the materials nowadays solely based on notorious and expensive aziridine. Based on the mechanism and achieved molecular masses, it may also allow access to new polyamine materials, not accessible from aziridine. The work seems well performed to me. The polymers are properly characterized and the detailed and well presented DFT calculations shed light on the possible mechanism of the reaction. Also, a lot of other alkanolamines were tested, the outcome well investigated and critically discussed. Therefore, i think this work has the potential to be suitable for the high standards of Nat. Comm. And to be also of interest for the broad readership of the journal. Nevertheless, certain point should be addressed in a revision, before acceptance can be recommended.

Abstract: I would take care in claiming a route "is greener" without any metric. For example, it is no known how the energy demand of this new route is compared to the stat-of-the-art routes and so on. Of course, the very toxic aziridine is replaced and I suggest using this as the selling point (which is a strong selling point). May just rewrite to "the reported method avoids the use of very toxic intermediates compared to the state-of-the-art" (current can be skipped).

Discussion on polymerization optimization: It is not clear to me, if the part before table 1 is also valid for the material made according to entry 1 or only for the one according entry 3? Are there also imine groups in the polymer according tom entry 1? If yes, please discuss. If the material according to entry 1 has no imine functions, are there any differences in color compared to the one made via entry 3? And if so, any explanation why imines are in them polymer with a anisole and THF but not toluene? Scale up and catalyst recyclability: It would be good to add one or two sentences or a simple scheme, how the recycling was performed. I assume the polymer was just filtered of (this is not even in the SI were this exp is described) after the first run and then the toluene solution was run again with fresh ethanolamine. The catalyst recycling is a very important result in such a polymerization were in principle a well coordinating and therefore metal scavenging material is formed, therefore it could be placed a bit more prominent.

The experiment with the COD hydrogenation did not add any relevant information. In the polyamine material made, besides imines there are enamine functions. Enamines have a significant different reactivity than simple alkenes (see also later in the discussion, please replace alkene by enamine). Therefore, a meaningful probe would just use a simple enamine instead of COD in an hydrogenation. For example the enamine formed from pyrrolidone and propanal or something similar (I 'm not sure, what is commercially available).

Substrate scope: Unfortunately, with most other aminialcohols, ring formation or amide formation occurs. But having a look in the ESI 1.9, two relevant aminoalcohols were no tested, which may give a polyamine: What happens if 1-amino-2-propanol is used? May it helps, if the NH₂-function is on the terminal position? The other aminioalcohol of interest is 6-amino-1-hexanol, as it would form a unfavored 7-memberd ring unlike the others who form preferred 5- and 6-memberd rings (to evaluate if this is the only reason why they are not working). I strongly suggest to test this two additional aminoalcohols (both are commercial available) and take the outcome into account

Reviewer #2:

Remarks to the Author:

In this paper, Brodie et al. report a method for the synthesis of ethoxylated branched polyethylenimine directly from ethanolamine using a Mn-pincer catalyst. The polymerization follows a hydrogen borrowing pathway to generate high molecular weight branched polyethylenimine. They used a range of characterization tools (multinuclear NMR and IR spectroscopy, GPC, TGA and DSC) to investigate the composition and the properties of the formed polymers and found a method to control the extent of nitrogen content in the final polymer. They performed in-depth DFT studies on the polymerization reaction to propose a catalytic cycle. The results are the extension of their previous work on the copolymerization of ethylene glycol and ethylenediamine using a similar strategy (Angew. Chem. Int. Ed., 2023, 62, e202306655). While the concept on itself is interesting, the overall quality of the experimental data questions some of the conclusions of the papers. I would not recommend considering this manuscript for publication until all the concerns below are solved.

Major concerns concerning the polymers:

1. The low resolution of the reported $^{13}\text{C}\{^1\text{H}\}$ NMR spectra does not allow to unambiguously assign the different linkages (potentially) present in all polymer samples. The convoluted IR spectra don't provide further support to the proposed assignments. Better resolution $^{13}\text{C}\{^1\text{H}\}$ NMR spectra should be recorded for all polymers. See comments 10 for further comments on the general presentation of the spectra.
2. Additionally, the paper only provides a very qualitative measure of the various linkages. The authors should quantify the degree of branching as well as providing an estimation of the different linkages (e.g. by quantitative $^{13}\text{C}\{^1\text{H}\}$ NMR or HSQC). These would considerably increase the quality of the discussion of the different catalytic runs as well as the subsequent mechanistic studies. The authors should also record $^{15}\text{N}\{^1\text{H}\}$ NMR spectroscopy (or $^1\text{H}\text{-}^{15}\text{N}$ HMBC to improve the resolution of the measurement) as an additional characterization tool.
3. The authors should do a quantitative analysis of the hydroxyl end-group by using a phosphorous-containing reagent such as 2-chloro-4,4,5,5-tetramethyl dioxaphospholane (e.g., Nat. Commun. 2019, 10, 2668) to further characterize the polymer formed.
4. The shape of the TGA curves either suggests that the polymer samples are not pure (e.g. containing remaining solvent, monomer or byproduct) or there are many chemical decomposition events happening upon heating polymers. Extracting Td5% is therefore unreliable and so is the interpretation of the melting temperature obtained from the DSC curves. The authors cannot rule out that the observed endothermic events observed in some DSCs are not coming from evaporation of decomposition products. This would also be consistent with the powder XRD results showing essentially amorphous polymers. Better TGA and DSC data are therefore essential for considering the paper for publication. Alternatively, the authors should extensively explain what chemical deterioration events (e.g. using TGA-MS) are happening upon heating and provide convincing data that the endothermic peaks observed in some of the DSCs are not evaporation of some decomposition products.
5. P3, the authors wrote the following statement: "We speculate this polymodal distribution may be the result of the presence of unsaturation within the polymer (u-PEI-1)." This statement is unclear. Are the authors saying that they have two populations of polymers (hydrogenated and non-hydrogenated)? If this is the case, they should provide additional data that support this claim.
6. The authors are proposing a chain-growth mechanism only based (experimentally) on the narrow PDI observed throughout the polymerization. They provide additional DFT calculation results to support their argument. The current experimental data is not sufficient to differentiate a chain-growth mechanism from a step-growth polymerization. The authors should at least perform kinetic studies of monomer consumption as well as investigating the relationship between molecular weight and monomer conversion. This data would provide additional experimental support for one mechanism or

the other.

Concerns about catalysis:

7. The authors should include a discussion of the reproducibility of these polymerization reactions. Only one entry of the table (Entry 17) explicitly says "average of 2 repeats".

8. The authors claims that water is the only by-product and their proposed approach is greener/less toxic than current industrial processes. Looking at the entire chemical reaction is essential to write such a claim. Besides water, an equivalent of KBr and tert-butanol are formed upon the activation of the catalyst. Furthermore it requires the preparation of $\text{MnBr}(\text{CO})_5$ (which traditionally involves bromine) and the ligand (also generation of inorganic salts). The authors should therefore nuance more their claim or be more explicit in what they mean by "green".

9. Interestingly, KOTBu by itself catalyze the formation of diethanolamine. I am curious if KOTBu is not playing an essential role in the branching formation during the polymerization reaction also when performed under optimal conditions. However, KOTBu is completely left out of the proposed mechanism. The authors should therefore investigate its influence on the selectivity of the polymerization. For instance, they should try to independently prepare Mn-2 (mixing 1/1 ratio of Mn-1 + KOTBu) and test it for polymerization without the addition of KOTBu. The authors should also compare the results obtained with those obtained from the recycling test to see if similar polymers are obtained.

General comments:

10. In general, the ESI is poorly organized. It was a real challenge to find data for a specific reaction and to navigate throughout the manuscript. The first problem is that the table in the manuscript and in the ESI are not following the same order. This should be fixed. All ^{13}C NMR and IR spectra should be properly labelled and readable. Also, some data for some polymers are missing (e.g. Entry 4). The HSQC spectrum in Fig. S65 is not in line with the rest of the data. The authors should comment on that. They should also properly label the axes, provide a full spectrum with a zoom on the desired region for this figure. The ^{31}P NMR spectra should show the integrations (and all the NMR spectra where quantitative results are extracted). Finally, the authors should carefully go through all the caption to correct all the small mistakes/ inconsistencies and cross link all figure numbers in the ESI with the manuscript.

11. In Fig. 1A the reaction of PEI with ethylene oxide would not lead to a polymer as drawn if the figure. The n+1, o+1 and p+1 notations are not consistent with the previous drawing. The structure of the polymer should therefore be corrected. In Fig. 1B. KOTBu is missing above the reaction arrow. The chemical equation should also be completely balanced as for the Fig. 1A.

12. In P1, the authors mention a market size of PEI and PEIE of around £400 million. They should specify if this corresponds to the annual market.

Reviewer #3:

Remarks to the Author:

The present manuscript describes a very interesting application of the Mn-pincer catalysed hydrogen borrowing mechanism to the polymerization of ethanolamine. The manuscript may thus be potentially of interest to a wide readership of Nature Communications. The following points should be addressed by the authors in any case before publication:

1) While the polymers produced are characterized sufficiently regarding their composition, it remains at this stage open whether they can also be used as materials for possible applications as mentioned in the introduction. Do the authors have any evidence, or can they comment on this point?

2) The borrowing hydrogen activity of Mn-1 and related Mn- and Ru-MACHO-type complexes has been studied for alcohols in some detail by the groups of Beller and Leitner in particular. Cross reference to this work seems to be very relevant for the reader in particular also for the DFT part. The state-of-the-art is not fully reflected in this context.

3) Is there any evidence that the dehydrogenation might occur not only at the alcohol, but also at the amine part of the ethanolamine? Is there H/D scrambling if ethanolamine is kept under a D₂ atmosphere in presence of Mn-1/base or Mn-2?

Reviewer #1 (Remarks to the Author):

Comment: *This manuscript reports a very interesting and new approach to obtain highly valuable and versatile ethoxylated polyamines. It uses a simple and common starting material (ethanolamine) in combination an earth-abundant metal (manganese) based catalyst and therefore this approach could have a high potential for later applications. I could allow access to the materials nowadays solely based on notorious and expensive aziridine. Based on the mechanism and achieved molecular masses, it may also allow access to new polyamine materials, not accessible from aziridine. The work seems well performed to me. The polymers are properly characterized and the detailed and well presented DFT calculations shed light on the possible mechanism of the reaction. Also, a lot of other alkanolamines were tested, the outcome well investigated and critically discussed. Therefore, i think this work has the potential to be suitable for the high standards of Nat. Comm. And to be also of interest for the broad readership of the journal. Nevertheless, certain point should be addressed in a revision, before acceptance can be recommended.*

Response: We thank reviewer 1 for these positive comments on our work.

Comment: *Abstract: I would take care in claiming a route “is greener” without any metric. For example, it is no known how the energy demand of this new route is compared to the state-of-the-art routes and so on. Of course, the very toxic aziridine is replaced and I suggest using this as the selling point (which is a strong selling point). May just rewrite to “the reported method avoids the use of very toxic intermediates compared to the state-of-the-art” (current can be skipped).*

Response: This is good point, and we agree with the reviewer here. We have amended the abstract accordingly.

Comment: *Discussion on polymerization optimization: It is not clear to me, if the part before table 1 is also valid for the material made according to entry 1 or only for the one according to entry 3? Are there also imine groups in the polymer according to entry 1? If yes, please discuss. If the material according to entry 1 has no imine functions, are there any differences in color compared to the one made via entry 3? And if so, any explanation why imines are in them polymer with a anisole and THF but not toluene?*

Response: We thank the reviewer for their comment. We have added “Entry 3” here to clarify that this is the polymer we are referring to here.

We have also added here a footnote (ref 28) to indicate the colour of the polymer and to increase transparency here as it is possible that there are small imine impurities remaining that are not spectroscopically observable but are indicated by the polymer colour. The ref 28

says : “While no clear spectroscopic evidence of the presence of imines can be found for Entry 1, the colour of the polymer (deep yellow/brown) suggests there may be low levels of unsaturation present. This colour is the same for all samples produced.”

On the reason as to why greater degrees of saturation are reached in toluene: we believe this is likely the result of inhibition to the hydrogenation process through solvent coordination (i.e. in the cases of THF, anisole). However, without evidence we have chosen to leave this hypothesis out of the manuscript but include it here for the reviewer’s interest.

Comment: *Scale up and catalyst recyclability: It would be good to add one or two sentences or a simple scheme, how the recycling was performed. I assume the polymer was just filtered of (this is not even in the SI were this exp is described) after the first run and then the toluene solution was run again with fresh ethanolamine. The catalyst recycling is a very important result in such a polymerization were in principle a well coordinating and therefore metal scavenging material is formed, therefore it could be placed a bit more prominent.*

Response: We thank the reviewer for their comment. We have clarified this in the ESI and added a simple figure (S90) to illustrate this. The following text has been added in the ESI:

“To determine the recyclability of the catalyst, a polymerisation was carried out under standard conditions in accordance with Table S1, Entry 1. At the end of the reaction, the water soluble polymer (that is solubilised and phase-separates into the liberated water) is isolated *via* decantation and the toluene portion of the reaction mixture along with any toluene-soluble organics was transferred into an ampoule containing fresh ethanolamine (2 mmol, 0.12 mL). This ethanolamine containing reaction vessel was sealed under argon and heated to 150 °C for 24 hours. After this time, the reaction was allowed to cool to room temperature and the resulting product was extracted into distilled water (5 mL). Water and other volatile components were removed under reduced pressure at 110 °C to yield **PEI-1** (55 mg, 64 %).”

Comment: *The experiment with the COD hydrogenation did not add any relevant information. In the polyamine material made, besides imines there are enamine functions. Enamines have a significant different reactivity than simple alkenes (see also later in the discussion, please replace alkene by enamine). Therefore, a meaningful probe would just use a simple enamine instead of COD in an hydrogenation. For example the enamine formed from pyrrolidone and propanal or something similar (I’m not sure, what is commercially available).*

Response: We thank the reviewer for their comment. We have carried out the experiment as suggested (using the enamine formed by butanal and pyrrolidine), however, whilst the hydrogenation reaction removed the enamine functionality, a complex mixture of unidentifiable products were observed in both NMR and GC-MS (SI Section 1.6.6). The following text has

been added “Similar tests were also carried out using enamine 1-(1-butenyl)pyrrolidine which showed full consumption of the enamine signal, however with a complex mixture of unidentifiable products obtained (ESI, Section S1.6.6).” to detail this reaction.

We have replaced alkene with enamine in the manuscript and SI.

Comment: *Substrate scope: Unfortunately, with most other aminoalcohols, ring formation or amide formation occurs. But having a look in the ESI 1.9, two relevant aminoalcohols were not tested, which may give a polyamine: What happens if 1-amino-2-propanol is used? May it help, if the NH₂-function is on the terminal position? The other aminoalcohol of interest is 6-amino-1-hexanol, as it would form a unfavoured 7-membered ring unlike the others who form preferred 5- and 6-membered rings (to evaluate if this is the only reason why they are not working). I strongly suggest to test these two additional aminoalcohols (both are commercial available) and take the outcome into account*

Response: We thank the reviewer for their comment. 1-amino-2-propanol was trialled and produced a complex mixture of products of unidentifiable products, including a polymer with a Mn of 12,300 g mol⁻¹ and PDI of 1.16. Similarly, 6-amino-1-hexanol also produced a complex mixture of products including a polymer with a Mn of 43,000 g mol⁻¹ and PDI of 1.74 (containing imine, amine and amide groups). These reactions however were low-yielding (38% and 21% respectively) with produced polymer alongside a wide mixture of oligomers and other low-molecular weight products, so would need further optimisation and analysis. Initial analysis of these products are detailed in SI section 1.12. Since this paper is primarily focused on the novel generation of PEI-1 and the mechanism by which this process operates rather than the production of new polymers and so this is outside the scope, but have included these initial findings in the SI. We thank the reviewer for the suggestion though and may follow this up in a future study.

Reviewer #2 (Remarks to the Author):

In this paper, Brodie et al. report a method for the synthesis of ethoxylated branched polyethylenimine directly from ethanolamine using a Mn-pincer catalyst. The polymerization follows a hydrogen borrowing pathway to generate high molecular weight branched polyethylenimine. They used a range of characterization tools (multinuclear NMR and IR spectroscopy, GPC, TGA and DSC) to investigate the composition and the properties of the formed polymers and found a method to control the extend of nitrogen content in the final polymer. They performed in-depth DFT studies on the polymerization reaction to propose a catalytic cycle. The results are the extension of their previous work on the copolymerization of ethylene glycol and ethylenediamine using a similar strategy (Angew. Chem. Int. Ed., 2023, 62, e202306655). While the concept on itself is interesting, the overall quality of the experimental data questions some of the conclusions of the papers. I would not recommend considering this manuscript for publication until all the concerns below are solved.

Response: We thank the reviewer for their nice summary and believe that we have addressed their concerns in the below responses.

Comment 1. *The low resolution of the reported $^{13}\text{C}\{^1\text{H}\}$ NMR spectra does not allow to unambiguously assign the different linkages (potentially) present in all polymer samples. The convoluted IR spectra don't provide further support to the proposed assignments. Better resolution $^{13}\text{C}\{^1\text{H}\}$ NMR spectra should be recorded for all polymers. See comments 10 for further comments on the general presentation of the spectra.*

Response: We also wish we had better resolution $^{13}\text{C}\{^1\text{H}\}$ NMR spectra, however, the solubility of these polymers does limit the resolution we are able to obtain. The solubility of the polymer was tested in a wide range of solvents, of which D_2O provided the best signal/noise ratio. These tests included $\text{d}^1\text{-TFA}$ (trifluoroacetic acid) which did show better solubility, however loss of M_n was observed upon a short time-frame ($>40,000$ to 579 g mol^{-1} in 2 minutes, see SI section 1.7 for details). We have already used a cross-polarisation method to enhance the $^{13}\text{C}\{^1\text{H}\}$ resolution (the data presented) and used longer than standard experiments for this. We have re-collected $^{13}\text{C}\{^1\text{H}\}$ NMR data and updated the manuscript with the best data we could obtain. We do concede that IR is not a very sensitive technique and that as our $^{13}\text{C}\{^1\text{H}\}$ resolution is limited we cannot rule out the presence of low concentrations of additional functional groups (for instance, residual imines), and we have clarified this as a footnote in ref. 28.

“While no clear spectroscopic evidence of the presene of imines can be found for Entry 1, the colour of the polymer (deep yellow/brown) suggests there may be low levels of unsaturation present. This colour is the same for all samples produced.”

Comment 2. *Additionally, the paper only provides a very qualitative measure of the various linkages. The authors should quantify the degree of branching as well as providing an estimation of the different linkages (e.g. by quantitative $^{13}\text{C}\{^1\text{H}\}$ NMR or HSQC). These would considerably increase the quality of the discussion of the different catalytic runs as well as the subsequent mechanistic studies. The authors should also record $^{15}\text{N}\{^1\text{H}\}$ NMR spectroscopy (or ^1H - ^{15}N HMBC to improve the resolution of the measurement) as an additional characterization tool.*

Response: We thank the reviewer for their comment. We have subsequently performed some quantitative $^{13}\text{C}\{^1\text{H}\}$ NMR experiments on our optimised polymer and on the product from the scaled-up reaction to quantify the degree of branching present (ESI section 1.10), and the results incorporated into the manuscript. “Crude quantification of % primary amine has also been carried out, through reactions to form Ruhemann’s purple with UV-vis spectrometry and through quantitative $^{13}\text{C}\{^1\text{H}\}$ NMR spectroscopy, which showed ca. 27% primary amine (ESI section 1.10).”

$^{15}\text{N}\{^1\text{H}\}$ NMR spectroscopy was trialled, however the data obtained was of poor value due to issues with cross-peak intensity, so is not included.

Comment 3. *The authors should do a quantitative analysis of the hydroxyl end-group by using a phosphorous-containing reagent such as 2-chloro-4,4,5,5-tetramethyl dioxaphospholane (e.g., Nat. Commun. 2019, 10, 2668) to further characterize the polymer formed.*

Response: We thank the reviewer for this useful suggestion. The experiment suggested here is used regularly within the field of lignin chemistry and uses quantitative $^{31}\text{P}\{^1\text{H}\}$ NMR and the reactivity between a chlorophosphine (ClPR_2) and alcohol (X-OH , where X here could be polymer) to generate the phosphitylation product, X-O-PR_2 , as a way to quantify alcohol content in the presence of an alcohol internal standard. We did try this experiment using the procedure reported in literature (see *Molecules*, 2023, 28, 7885). However, it appears that some N-H reactivity was also observed (forming X-N-PR_2 type species). Indeed, this has been reported for these types of amines before (e.g., Sur Quelques Dérivés Phosphores de la Spermine, *Phosphorus, Sulfur, and Silicon and the Related Elements*, 1996, 115:1, 39-42). This unwanted reactivity has obfuscated the quantification of hydroxy groups using this method as both alcohol and amine phosphitylation products appear in a similar region within the resulting $^{31}\text{P}\{^1\text{H}\}$ NMR spectrum.

Comment 4. *The shape of the TGA curves either suggests that the polymer samples are not pure (e.g. containing remaining solvent, monomer or byproduct) or there are many chemical decomposition events happening upon heating polymers. Extracting Td5% is therefore*

unreliable and so is the interpretation of the melting temperature obtained from the DSC curves. The authors cannot rule out that the observed endothermic events observed in some DSCs are not coming from evaporation of decomposition products. This would also be consistent with the powder XRD results showing essentially amorphous polymers. Better TGA and DSC data are therefore essential for considering the paper for publication. Alternatively, the authors should extensively explain what chemical deterioration events (e.g. using TGA-MS) are happening upon heating and provide convincing data that the endothermic peaks observed in some of the DSCs are not evaporation of some decomposition products.

Response: We agree with the reviewers point here. We did make a concerted effort to dry these polymers before analyses (under reduced pressure at 10^{-3} mbar at 110 °C for several hours before all analyses to remove any residual solvents/substrate). TGA-MS analysis was run on sample S1 Entry 11 due to exhibiting strong deterioration signals. Whilst this showed similar decomposition, the ion count was insufficient to distinguish unique signals against background noise. A blank control sample of ethanolamine was also ran, which showed the same deterioration onset in the TGA, further implying that the initial loss could be due to residual starting material (Figures S69-S71, ESI).

We do agree that it is difficult to ascertain an accurate decomp. temperature from these complex materials though – and so we have amended the manuscript to remove these 5% mass loss values and have placed a more general statement as in the below:

“Complex thermal decomposition with significant mass loss is observed above 200 °C, with some small mass loss below this temperature which could indicate the presence of residual substrate/impurities within the samples despite drying under reduced pressure and elevated temperatures prior to analyses. Performing powder X-ray diffraction on a sample of PEI-1 reveals this material to be largely amorphous.”

Comment 5. *P3, the authors wrote the following statement: “We speculate this polymodal distribution may be the result of the presence of unsaturation within the polymer (u-PEI-1).” This statement is unclear. Are the authors saying that they have two populations of polymers (hydrogenated and non-hydrogenated)? If this is the case, they should provide additional data that support this claim.*

Response: We thank the reviewer for their comment. This a fair point – we have removed this ambiguous sentence. The cases where this polymodal distribution occurs are during the optimisation, where unsaturation was observed using NMR and IR spectroscopy. These data are provided in the ESI. GPC data is indicative of two polymers for which there are multiple possibilities, hydrogenated and non-hydrogenated being one of them. Other possibility could be two polymers containing different arrangements/branching of primary, secondary, and

tertiary amines or imines. It is not easily possible to separate these polymers and identify them and we cannot be certain about the identity of two polymers based on the spectroscopic data. It is worth noting that the polymodal distribution is only observed in some cases when reaction conditions are varied, and our optimised condition leads to the formation of just one polymer (as confirmed by the GPC).

Comment 6. *The authors are proposing a chain-growth mechanism only based (experimentally) on the narrow PDI observed throughout the polymerization. They provide additional DFT calculation results to support their argument. The current experimental data is not sufficient to differentiate a chain-growth mechanism from a step-growth polymerization. The authors should at least perform kinetic studies of monomer consumption as well as investigating the relationship between molecular weight and monomer conversion. This data would provide additional experimental support for one mechanism or the other.*

Response: We thank the reviewer for their comment. We agree that proposing chain growth mechanism will need more detailed kinetic data.

Unfortunately, monomer consumption is not easily monitored in this case. Here, we have a H₂ borrowing process which requires the vessel to remain sealed. This would introduce the first error to a possible measurement as we would need to run these as many experiments in parallel rather than continuous monitoring of one single experiment. Then we come to the issue of how to measure our monomer concentration at different time points – due to our, proposed, chain growth mechanism there is polymer formation of high molecular weight early on in the reaction. This results in complex NMR spectra (see section S1.4.3) where monomer and polymer signals are overlapping – meaning no meaningful integration/quantification can be obtained through NMR. Additionally, the presence of the polymer early on means GC quantification is also not an option for monitoring these reactions.

As mentioned in the manuscript, the DFT computation doesn't show any preference towards chain growth or step growth. We have therefore added a statement: "Thus, based on our studies we are unable to confirm whether the chain propagation occurs via chain-growth or step-growth and it is possible that both the pathways are occurring under the reaction conditions."

Comment 7. *The authors should include a discussion of the reproducibility of these polymerization reactions. Only one entry of the table (Entry 17) explicitly says "average of 2 repeats".*

Response: The optimised conditions polymerisation experiment is very reproducible – we have done this experiment many times in our laboratory. We do not think we need to run

repeats of the non-optimised conditions as these are not as important and simply demonstrate our procedure for making the purest sample of PEI-1 in high yield. The manuscript has been updated as follows to highlight the reproducibility of these polymerisations.

“Remarkably, refluxing ethanolamine (2 mmol) in toluene (4 mL) using 1 mol% complex Mn-1, 10 mol% K₂OtBu at 150 °C for 24 h in a sealed vessel resulted in the isolation of a solid product in 81% yield (Table 1; Entry 1; reproduced three times).”

Comment 8. *The authors claims that water is the only by-product and their proposed approach is greener/less toxic than current industrial processes. Looking at the entire chemical reaction is essential to write such a claim. Besides water, an equivalent of KBr and tert-butanol are formed upon the activation of the catalyst. Furthermore it requires the preparation of MnBr(CO)₅ (which traditionally involves bromine) and the ligand (also generation of inorganic salts). The authors should therefore nuance more their claim or be more explicit in what they mean by “green”.*

Response: We thank the reviewer for their comment. We have toned this down in accordance with the reviewer response. We have made changes in abstract by replacing “ this method is greener” with “this method avoids toxic intermediate”.

Although we will point out here that it is only 1 equivalent of KBr and tBuOH with respect to the catalyst – not the substrate – so we are sub-stoichiometric equivalence here. We have also demonstrated the recyclability of our catalyst, meaning these side-products would only be generated in the initial catalyst activation and not in further batches for instance.

Comment 9. *Interestingly, K₂OtBu by itself catalyze the formation of diethanolamine. I am curious if K₂OtBu is not playing an essential role in the branching formation during the polymerization reaction also when performed under optimal conditions. However, K₂OtBu is completely left out of the proposed mechanism. The authors should therefore investigate its influence on the selectivity of the polymerization. For instance, they should try to independently prepare Mn-2 (mixing 1/1 ratio of Mn-1 + K₂OtBu) and test it for polymerization without the addition of K₂OtBu. The authors should also compare the results obtained with those obtained from the recycling test to see if similar polymers are obtained.*

Response: This is a good idea and one we also had. The experiment using **Mn-2** (the activated complex) has been run as advised which resulted in the formation of a polymer with similar yields to the optimised procedure (77% and 81% respectively), and with similar Mn and PDI values (M_n 33,200 g mol⁻¹, Đ 1.29 compared with 38,700 g mol⁻¹, Đ 1.21). However, the ¹³C{¹H} NMR spectrum showed significant difference with a very broad region for the secondary amine region, with a lower % of primary amines, and higher % of 3° amines,

implying a more cross-linked polymer. This result has now been included in the manuscript as follows and incorporated into the SI (Section 1.6.8).

“Performing the reaction with pre-activated **Mn-2** and the absence of base also yields a polymer in a similar yield (77%) (Table 1, Entry 17), the degree of branching is significantly higher, with only ca. 5% primary amine, with a greater proportion of tertiary amine (see ESI, Sections 1.6.9 and 1.10.2).”

Comment 10. *In general, the ESI is poorly organized. It was a real challenge to find data for a specific reaction and to navigate throughout the manuscript. The first problem is that the table in the manuscript and in the ESI are not following the same order. This should be fixed. All ¹³C NMR and IR spectra should be properly labelled and readable*

Response: We believe our ESI is easy to navigate. A reference table is provided along with data summaries (including all shifts) and all spectra are presented in clear subsections categorised by analyses type. The reason why the tables in the manuscript and SI are different is due to the SI table focused on optimisation, with control reactions in later sections in greater detail. In contrast, the manuscript table is for all key polymerisation results discussed chronologically throughout the paper. As such, there are certain results excluded from each, which would make an identical table not possible. The signals observed in the IR spectra have been included onto the figures as advised.

Comment: *Also, some data for some polymers are missing (e.g. Entry 4).*

Response: Unfortunately, due to the low yields obtained within the optimisation of the reaction, full characterisation was not possible for all the polymers. (We have noted this on page S8: “Due to the low yield obtained for this reaction, infrared spectroscopy and thermal analysis were not carried out.”

Comment: *The HSQC spectrum in Fig. S65 is not in line with the rest of the data. The authors should comment on that. They should also properly label the axes, provide a full spectrum with a zoom on the desired region for this figure.*

Response: We thank the reviewer for their comment. A new HSQC experiment has been included in the ESI to replace this spectrum, with zoom in on desired region as requested by the reviewer (Figure S57).

Comment: *The ³¹P NMR spectra should show the integrations (and all the NMR spectra where quantitative results are extracted).*

Response: We thank the reviewer for their comment. This has now been done.

Comment: Finally, the authors should carefully go through all the caption to correct all the small mistakes/ inconsistencies and cross link all figure numbers in the ESI with the manuscript.

Response: We thank the reviewer for their comment. This has now been done.

Comment 11. In Fig. 1A the reaction of PEI with ethylene oxide would not lead to a polymer as drawn in the figure. The $n+1$, $o+1$ and $p+1$ notations are not consistent with the previous drawing. The structure of the polymer should therefore be corrected.

Response: We thank the reviewer for their comment. This has been amended in the manuscript to show more clearly the secondary reaction with ethylene oxide.

Comment: In Fig. 1B. KOtBu is missing above the reaction arrow. The chemical equation should also be completely balanced as for the Fig. 1A.

Response: We thank the reviewer for their comment. This has been added to the figure.

Comment 12. In P1, the authors mention a market size of PEI and PEIE of around £400 million. They should specify if this corresponds to the annual market.

Response: We thank the reviewer for their comment. This is annual market, amended in the manuscript.

Reviewer #3 (Remarks to the Author):

The present manuscript describes a very interesting application of the Mn-pincer catalysed hydrogen borrowing mechanism to the polymerization of ethanolamine. The manuscript may thus be potentially of interest to a wide readership of Nature Communications. The following points should be addressed by the authors in any case before publication:

We thank reviewer 3 for their positive comments on our work.

Comment 1) *While the polymers produced are characterized sufficiently regarding their composition, it remains at this stage open whether they can also be used as materials for possible applications as mentioned in the introduction. Do the authors have any evidence, or can they comment on this point?*

Response: We thank the reviewer for their comment. This is a broad question of course and relates to the applications of the new materials produced here, rather than on their synthesis (which is the topic of this manuscript). We have begun studies which show promising results in a variety of applications, including some of those mentioned in the introduction. However, these are follow-up studies that are out with the scope of this manuscript and we plan to report these as stand-alone manuscripts allowing for an in-depth investigation.

Comment 2) *The borrowing hydrogen activity of Mn-1 and related Mn- and Ru-MACHO-type complexes has been studied for alcohols in some detail by the groups of Beller and Leitner in particular. Cross reference to this work seems to be very relevant for the reader in particular also for the DFT part. The state-of-the art is not fully reflected in this context.*

Response: We thank the reviewer for their comment. We have added some additional references from these named groups to our manuscript as we agree they are important players in the field:

37. Elangovan, S. *et al.* Efficient and selective N-alkylation of amines with alcohols catalysed by manganese pincer complexes. *Nat. Commun.* **7**, 1–8 (2016).
38. Kaithal, A., Gracia, L. Lou, Camp, C., Quadrelli, E. A. & Leitner, W. Direct Synthesis of Cycloalkanes from Diols and Secondary Alcohols or Ketones Using a Homogeneous Manganese Catalyst. *J. Am. Chem. Soc.* **141**, 17487–17492 (2019).
39. Kaithal, A., van Bonn, P., Hölscher, M. & Leitner, W. Manganese(I) Catalyzed Methylation of Alcohols Using Methanol as C1 Source. *Angew. Chem. Int. Ed.* **59**, 215–220 (2020).

Comment 3) *Is there any evidence that the dehydrogenation might occur not only at the alcohol, but also at the amine part of the ethanolamine? Is there H/D scrambling if ethanolamine is kept under a D2 atmosphere in presence of Mn-1/base or Mn-2?*

Response: We thank the reviewer for their comment. A control experiment where hexylamine was used instead of ethanolamine under the same reaction condition showed no conversion of hexylamine suggesting that it is unlikely that amine will be dehydrogenated under the reaction conditions. Additionally, from our DFT computation we do not believe that amine dehydrogenation would be favoured here. The barrier for the hydrogenation of the imine dimer is 21.4 kcal/mol (from Mn-4 to TS-6) and strongly exergonic (-11.4 kcal/mol, Figure S180). The reverse would be the dehydrogenation of the secondary amine (5), with a barrier on this profile of 36.4 kcal/mol which is significantly higher. To compare, the alcohol dehydrogenation of the dimer (Figure 3C) has a barrier of 17.1 kcal/mol, significantly lower than the dehydrogenation of the secondary amine.

We have carried out the polymerisation of ethanolamine under a D₂ atmosphere and deuterium incorporation is observed into the polymer backbone. This has been described in the SI Section 1.6.8. Based on computational work, dehydrogenation would be followed by subsequent polymerisation and hydrogenation, so H/D scrambling is supportive of hydrogen-borrowing mechanism as described in the manuscript. This has been discussed in the manuscript.

Reviewers' Comments:

Reviewer #1:

Remarks to the Author:

In this revised manuscript the authors addressed my comments on the initial submission very well as well as the ones from the other reviewers (at least from my perspective). The structure of the Supporting Information is fine. But I still recommend to skip the COD hydrogenation experiments because as already commented, this did not add any relevant information for this system. Just skip the COD part in the manuscript (lines 211-215) as well as the corresponding information in the ESI (lines 733-761). The observations on the Imine as well as enamine hydrogenation are sufficient. After addressing this, the manuscript is suitable to be recommended for publication in Nature Communications.

Reviewer #2:

Remarks to the Author:

The authors have addressed all my concerns and I recommend this manuscript to be published in Nature Communications in the current form.

Reviewer #3:

Remarks to the Author:

The authors have provided extensive discussion of all reviewer comments and added additional data and discussions in the text and SI, improving the quality of the manuscript and answering many of the questions explicitly.

For the question regarding the catalytic reaction, their revisions are addressing all points in a satisfactory manner. I am only wondering whether they may want to include the observation about lack of H/D exchange in the amine functionality and D-incorporation in the polymer with a short comment at the mechanistic discussion in the paper (it seems to appear only in the response to the reviewers).

A significant amount of additional data on polymer characterization and relevant control experiments has also been added, improving the manuscript in this respect. It seems that not all suggestions could be implemented, but this appears to be due mainly to inherent limitations of the methods or materials rather than leaving major concerns about the conclusions. However, I would leave it to the polymer chemist experts as reviewers to finally approve that this has addressed all possible open points.

In essence, I can support acceptance of the manuscript from the perspective of catalysis.

Reviewer #1 (Remarks to the Author):

In this revised manuscript the authors addressed my comments on the initial submission very well as well as the ones from the other reviewers (at least from my perspective). The structure of the Supporting Information is fine. But I still recommend to skip the COD hydrogenation experiments because as already commented, this did not add any relevant information for this system. Just skip the COD part in the manuscript (lines 211-215) as well as the corresponding information in the ESI (lines 733-761). The observations on the Imine as well as enamine hydrogenation are sufficient. After addressing this, the manuscript is suitable to be recommended for publication in Nature Communications.

Response: We thank the reviewer for their time to evaluate the manuscript and their comment. We take the point that since enamines and not alkenes are proposed intermediates in the reported polymerisation process, details on the hydrogenation of COD are less relevant. However, as mentioned in the paper and the ESI, the experiment on the hydrogenation of an enamine [1-(1-butenyl)pyrrolidine] was inconclusive under our reaction conditions. Since making a model enamine intermediate (e.g. 7-B) in Figure 4 would be challenging, we believe that the fact that the catalyst can hydrogenate an alkene is still useful for our mechanistic proposal as technically an enamine is an alkene with an amine substituent. We therefore prefer to leave it as it is.

Reviewer #2 (Remarks to the Author):

The authors have addressed all my concerns and I recommend this manuscript to be published in Nature Communications in the current form.

Response: We thank the reviewer for their time and considering our manuscript worthy of publication.

Reviewer #3 (Remarks to the Author):

The authors have provided extensive discussion of all reviewer comments and added additional data and discussions in the text and SI, improving the quality of the manuscript and answering many of the questions explicitly.

For the question regarding the catalytic reaction, their revisions are addressing all points in a satisfactory manner. I am only wondering whether they may want to include the observation about lack of H/D exchange in the amine functionality and D-incorporation in the polymer with a short comment at the mechanistic discussion in the paper (it seems to appear only in the response to the reviewers).

A significant amount of additional data on polymer characterization and relevant control experiments has also been added, improving the manuscript in this respect. It seems that not all suggestions could be implemented, but this appears to be due mainly to inherent limitations of the methods or materials rather than leaving major concerns about the conclusions. However, I would leave it to the polymer chemist experts as reviewers to finally approve that this has addressed all possible open points.

In essence, I can support acceptance of the manuscript from the perspective of catalysis.

Response: We thank the reviewer for their time to evaluate the manuscript and their comments. The details regarding H/D exchange are mentioned at the top of page 10.